# Antibiotic consumption study in two hospitals in Asmara from 2014 to 2018 using WHO's defined daily dose (DDD) methodology

**Nebyu Daniel Amaha** [1]*, **Dawit G. Weldemariam** [2], **Yohana H. Berhe** [3]

**1** Department of Nutrition and Dietetics, College of Health Sciences, Mekelle University, Mekelle, Ethiopia, **2** Department of Pharmacy, Hazhaz Hospital, Asmara, Eritrea, **3** Pharmacy, Addis Ababa, Ethiopia

* nebyudan@gmail.com

**Data Availability Statement:** All relevant data are within the manuscript and its Supporting Information files.

**Funding:** The author(s) received no specific funding for this work.

## Abstract

This study compares the antibiotic consumption rates over a period of five years in two hospitals in Eritrea, Orotta National Referral and Teaching Hospital (ONRTH) and Hazhaz Zonal Referral Hospital (HZRH). Antibiotic consumption is an important parameter in the study of antibiotic use. However, no published data on consumption rates exist for these two hospitals, thus the aim of the study is to measure and compare a five year antibiotic consumption trends of the two hospitals' medical wards using Defined Daily Dose per 100 bed-days (DDD/100-BD). Antibiotics dispensed from January 2014 to December 2018 were considered. Non-parametric Pearson's correlation coefficient was used for comparison of consumption, while non-parametric Friedman's test measured annual rates. The total antibiotic consumption in the HZRH was almost double that of ONRTH. The analysis showed that antibiotic consumption in ONRTH was significantly decreased from 2014 to 2018 while there was no significant difference in consumption in HZRH. Benzyl penicillin was the most consumed antibiotic in HZRH and ONRTH throughout the study period at 87.8DDD/100-BD and 35.4 DDD/100-BD respectively. Ceftriaxone and ciprofloxacin were among the most commonly consumed antibiotics in both hospitals. Establishment of Antibiotic stewardship program would benefit both hospitals greatly, and further studies need to be done to establish the national antibiotic consumption baseline.

## Introduction

Overuse and misuse of antibiotics are among the leading factors in the complex web of causation in antimicrobial resistance (AMR). The wide use of antibiotics, whether appropriate or not, exerts a selective pressure by reducing the reproductive success of some microorganisms thereby accelerating the development of AMR [1–4]. The amount of antibiotics prescribed, the number of patients treated with the antibiotics and the proportion of patients on antibiotics in a hospital are three important factors affecting this selection pressure [4]. Antibiotic consumption is rising globally, especially in low and middle-income countries [5]. Studies have shown that the choice of antibiotic or its duration is incorrect in 25% to 75% of cases. Irrational use of

**Competing interests:** The author Y.H.B. has a commercial affiliation with Pharmacy, Addis Ababa. The commercial affiliation did not provide financial support in the form of author salary to Y.H.B. D.G. W. was working in Hazhaz Zonal Referral Hospital at the time of submission of this manuscript. This does not alter our adherence to PLOS ONE policies on sharing data and materials.

antibiotics include prescribing antibiotics for infections of viral origin, using the wrong type of antibiotic, using the wrong dose, duration or route of administration, increased use of antibiotics in agriculture and the frequent use of broad-spectrum and last-resort antibiotics. There is a clear association between antibiotic consumption and the emergence of resistant microorganisms [6–8]. Therefore, reducing the consumption of unnecessary antibiotics would lower the antibiotic resistance rate and lessen the higher healthcare costs associated with drug resistant infections [1, 9–12].

The increasing resistance of bacteria to antibiotics has been pointed out as one of the main public health concerns globally, as treatment for growing list of infections becomes less effective and previously relatively safe medical procedures like surgery and organ transplants would become increasingly risky [1]. AMR increases healthcare expenditure, jeopardizes the achievement of Sustainable Development Goals and its impact disproportionately affects low and middle-income countries. And with the increase in consumption of antibiotics worldwide, in the USA least one of the commonly prescribed antibiotics shows resistance in more than 70% of bacterial infections which occur in hospitals [13].

Antibiotic consumption is an important parameter in the study of antibiotic use and there are several metrics of measuring antibiotic consumption. The most commonly used method is the one recommended and updated by the WHO Collaborating Centre for Drug Statistics Methodology World Health Organization (WHO) [14]. It is based on the concept of defined daily doses (DDD). The WHO global methodology is based on the Anatomical Therapeutic Chemical (ATC) and it classifies antibiotics' pharmacologically active substance based on the organ or system on which they act and on their therapeutic, pharmacological and chemical properties. Antibiotics are prescribed in different unit dose of daily administration, and hence a standard method should be used to measure antibiotic consumption. The DDD is the assumed average maintenance dose per day of an antibiotic substance(s) used for its main indication in adults, and is assigned to active ingredients with an existing ATC code. The ATC/DDD methodology was developed to improve the quality of patient care through research and monitoring the consumption of antibiotics [10, 15–17].

Expressing antibiotic consumption in DDD per 100 patient-days (DDD/100-BD) allows hospitals to compare their consumption with other hospitals regardless of differences in quality and quantity of antibiotics [3, 18]. No data has previously been published regarding the antibiotic consumption rates in Eritrean hospitals or wards, thus the amount and types of antibiotics consumed is unknown. The aim of this study was to measure and compare five year antibiotic consumption trends in the medical wards of two hospitals, Orotta National Referral Teaching Hospital (ONRTH) and Hazhaz Zonal Referral Hospital (HZRH) using the ATC/DDD methodology.

## Materials and methods

### Study design and setting

A five-year retrospective study which focused on the antibiotics dispensed to the internal medicine wards of two hospitals in Asmara was conducted, i.e. in Orotta National Referral and Teaching Hospital (ONRTH) and Hazhaz Zonal Referral Hospital (HZRH). Eritrea has six administrative regions and the central region is the main hub of the country. HZRH is a regional referral hospital for the central region. These two hospitals are found in Asmara, the capital and the most populated city in the country, located in the central region. Eritrea has two national referral hospitals and currently no privately owned hospitals are operational in the country. ONRTH is one of the two national referral hospitals in the country with a functional Medicine and Therapeutics Committee (MTC) while HZRH is a secondary level zonal

referral hospital which has established an MTC in 2019. ONRTH is a multi-specialty national referral hospital with medical, surgical, pediatric, emergency, intensive care unit (ICU), gynecology and obstetrics wards while HZRH is a secondary level zonal referral hospital with medical, pediatric and emergency wards.

### Antibiotic inclusion criteria

Antibiotics which were classified as J01 category (antibiotics for systemic use) under the ATC classification system and antibiotics available in the latest edition of the Eritrean National List of Medicines (ENLM) 2015 were included in this study [19]. Amount of antibiotics dispensed for 60 consecutive months i.e. from January 2014 up to December 2018 were consulted and data extracted from pharmacy records using a table with antibiotic's name, strength, amount dispensed and date dispensed. The following fourteen antibiotics which satisfied both conditions were included in the study (Table 1). The WHO in 2017 suggested that antibiotics be classified into three groups; Access, Watch or Reserve (AWaRe) groups and we used the 2019 database to classify them accordingly [20]. Ten antibiotics were in the Access group, four antibiotics in the Watch group and none were in the Reserve group (Table 1).

### Calculation of antibiotic consumption rate

The DDD is a globally accepted unit of measuring drug consumption of different strengths, pack sizes or combinations. It can be used to compare rates between regions, countries, hospitals and wards [21–24]. Number of DDDs was calculated by first converting the total amount of antibiotic dispensed in a given year into grams; this was then divided by the standard WHO DDD value given in grams (Refer the WHO site. Bed-days (BD) is given by multiplying three variables the number of beds, the bed occupancy rate and the number of days in the study. When measuring antibiotic consumption in an inpatient setting DDD/100-BD is the recommended method [25]. DDD/100-BD is given by dividing the number of DDDs by patient-days and multiplied by 100. To compare changes in consumption among the antibiotics we

**Table 1. Antibiotics for systemic use included in this study.**

| Antibiotic | ATC Code | Route | WHO DDD [14] | AWaRe Group |
|---|---|---|---|---|
| Amoxicillin | J01CA04 | O | 1.5 | Access |
| Ampicillin | J01CA01 | P | 6 | Access |
| Benzyl Penicillin G | J01CE01 | P | 3.6 | Access |
| Ceftriaxone | J01DD04 | P | 2 | Watch |
| Chloramphenicol | J01BA01 | P | 3 | Access |
| Ciprofloxacin | J01MA02 | O | 1 | Watch |
| Clarithromycin | J01FA01 | O | 0.5 | Watch |
| Cloxacillin | J01CF02 | P | 2 | Access |
| Co-trimoxazole | J01EE01 | O | 4 UD† | Access |
| Doxycycline | J01AA02 | O | 0.1 | Access |
| Flucloxacillin | J01CF05 | O | 2 | Access |
| Gentamycin | J01GB03 | P | 0.24 | Access |
| Metronidazole | J01XD01 | P | 1.5 | Access |
| Streptomycin | J01GA01 | P | 1 | Watch |

*P: Parenteral, O: Oral

† Unit Dose; AWaRe = Access, Watch, Reserve.

calculated the percent contribution of each antibiotic to the total antibiotic consumption of that year and scale it to 100, to enable us to see the "consumption percentage" of that antibiotic.

## Statistical analysis

Antibiotic consumption data were aggregated at the fifth (chemical substance) level of the ATC classification and expressed in DDD and DDD/100-BD. Data were entered and analyzed using Microsoft Excel ® 2010 and after data cleaning, it was exported to IBM SPSS ® version 23 for descriptive and analytical studies. Non-parametric Pearson's correlation coefficient was used to test for comparison of rates between the two hospitals. While annual change in antibiotic consumption was tested using nonparametric Friedman's test. All statistical tests were considered significant at p-value of $< 0.01$.

## Results

### Total antibiotic consumption

Total antibiotic consumption differed between the two hospital wards. Antibiotic consumption was consistently higher in Hazhaz Medical Ward (HMW) than in Orotta Medical Ward (OMW). During the five years, the consumption in HMW showed considerable variation, while there was relatively small variation in OMW (Fig 1).

The mean antibiotic consumption rate in DDD/100-BD versus year graph shows that antibiotic consumption rate was much higher in HMW than OMW throughout the five years. The

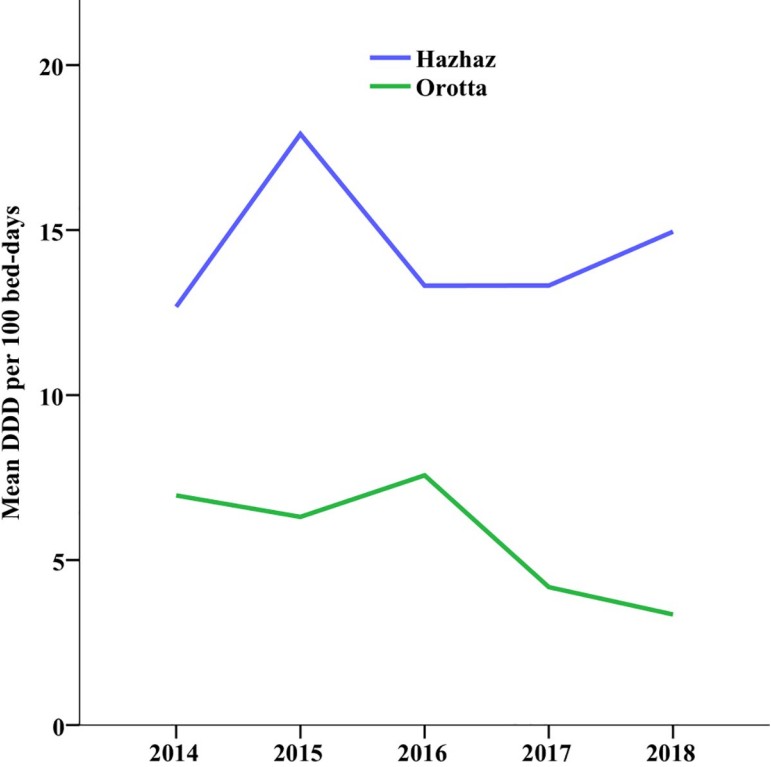

**Fig 1. Antibiotic consumption rate in Orotta and Hazhaz medical wards from 2014 to 2018 in mean DDD per 100 bed-days.**

consumption rates peaked in HMW in 2015 and fell down in 2016 and started to slowly increase thereafter, the rate of increase was higher in 2017 but it never approached the peak seen in 2015. In OMW, on the other hand, the line graph of antibiotic consumption shows less fluctuations and a small peak was observed in 2016 after which the consumption rate started to decrease consistently exhibiting the lowest rate in 2018 (Fig 1).

Consumption of antibiotics was not correlated between the two hospitals ($r$ = 0.151; $p$ = 0.93). Differences in annual antibiotic consumption rates were significant in OMW for the five years under study $\chi 2$ (4, N = 14) = 24.487, p<0.0001; while the annual variations in HMW were not significant $\chi 2$ (4, N = 14) = 2.86, p = 5.82. Follow-up pairwise comparisons were conducted using Wilcoxon test and controlling for Type I errors across these comparisons at the 0.005 level using Bonferroni procedure. The mean antibiotic consumption difference between 2014 and 2018 and between 2016 and 2018 were significant at p-values of 0.001 and 0.002 respectively. The remaining eight different comparisons were not significant. Consumption of antibiotics was not correlated between the two hospitals (r = 0.151; p = 0.93).

### Trends in antibiotic consumption

ONRTH is a national referral hospital and the number of beds in the medical ward has more than doubled between 2014 and 2018 while the number of beds in HMW has stayed relatively stable. The number of bed-days has shown a similar upward pattern in OMW, more than tripling from 8303 bed-days in 2014 to more than 30,500 patient-days in 2018 (Table 1).

Parenteral route of administration is more commonly used than the oral route in both hospitals. Oral antibiotics consumption showed a decrease overtime whereas parenteral antibiotic consumption decreased after peaking in 2015 but again started to increase in 2018 (Fig 2).

Benyzl penicillin G, gentamycin, amoxicillin, ciprofloxacin and ceftriaxone were the top five consumed antibiotics in OMW and HMW. Benzyl penicillin G showed an increase in consumption percentage from 2014 to 2018, and ceftriaxone's consumption percentage decreased in 2015 from its baseline and increased from 2016 to 2018. Amoxicillin showed a major decrease in consumption percentage in 2017 after it was steadily increasing from 2014 to 2016. Gentamycin and ceftriaxone exhibited a similar trend of increasing in consumption percentage (Fig 3).

The two wards showed difference in their choice of antibiotics and the amounts they dispense (Table 3). Benzyl penicillin G was consistently the most commonly consumed drug from 2014–2018 in both hospitals followed by gentamycin, amoxicillin and ciprofloxacin. The DDD/100-BD value of gentamycin was 17.7 in HMW which was almost 4 times than what was observed in OMW at 3.7 DDD/100-BD. Amoxicillin was the third highest consumed antibiotic in HMW; this antibiotic showed an erratic consumption patterns in OMW with a mean DDD/100-BD of 4.34 and standard deviation (SD) of 4.01, while in HMW it was almost six times that with a mean of 30.46 and SD 30.94. While gentamycin was the second antibiotic of choice in HMW, in OMW ciprofloxacin was second most commonly consumed antibiotic with a mean 7.6 DDD/100-BD.

### Discussion

This study found out that antibiotic consumption was higher in the secondary level hospital's ward than in the tertiary and this is in contrast with a study from New Zealand in which the consumption of antibiotic was higher in a tertiary hospital than in a secondary one. The total antibiotic consumption in HMW (158.5 DDD/100-BD) was higher than the 117.6 DDD/100-BD reported from a similar secondary level hospital study in New Zealand [26]. The mean antibiotic consumption in OMW was 79.5 DDD/100-BD lower from the tertiary medical

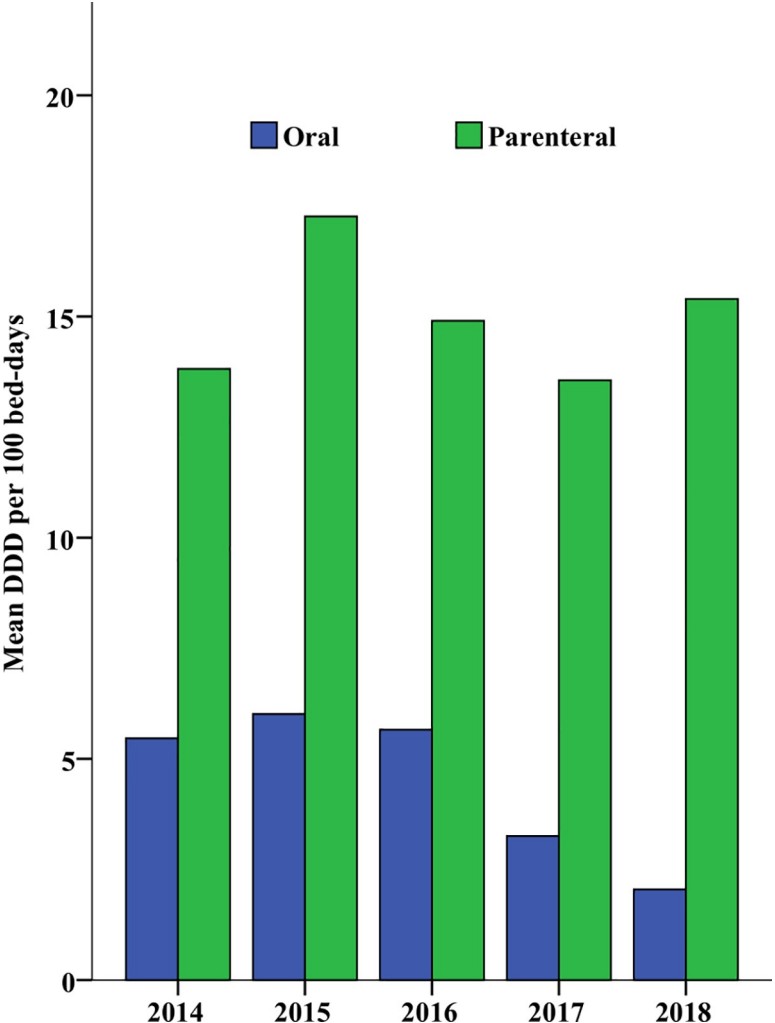

**Fig 2. Route of administration of antibiotics consumed in Orotta and Hazhaz medical wards from 2014 to 2018 given in mean DDD per 100 bed-days.**

wards in Ethiopia 91.8 DDD/100-BD [27] and Zurich 110.4 DDD/100-BD [18]. However, antibiotic consumption in OMW was higher than the 72.3 DDD/100-BD reported in Netherlands [23]. The difference in antibiotic consumption between Orotta and Hazhaz hospitals might be explained by differences in patients' characteristics which get referred to the zonal and national referral hospitals. DDD methodology does not take into account the reduction of doses in renal and hepatic failure patients and in case of a national referral hospital with many such patients, DDD would underestimate the amount of antibiotic consumed. Antibiotic policies or prescribers' specialty level might be some additional factors which could account for the difference between the two hospitals. We can observe that the number of beds has more than doubled in OMW between 2014 and 2018 whilst it remained constant in HMW (Table 2).

Antibiotic consumption showed opposite trends, decreasing in OMW and increasing in HMW (Fig 1). It decreased in OMW from 97.42 DDD/100-BD in 2014 to 46.91 DDD/100-BD in 2018, a similar drop was observed in medical ward of a teaching hospital in Turkey from 76 DDD/100-BD to 51.8 DDD/100-BD between 2011 and 2012 [8]. Antibiotic consumption in

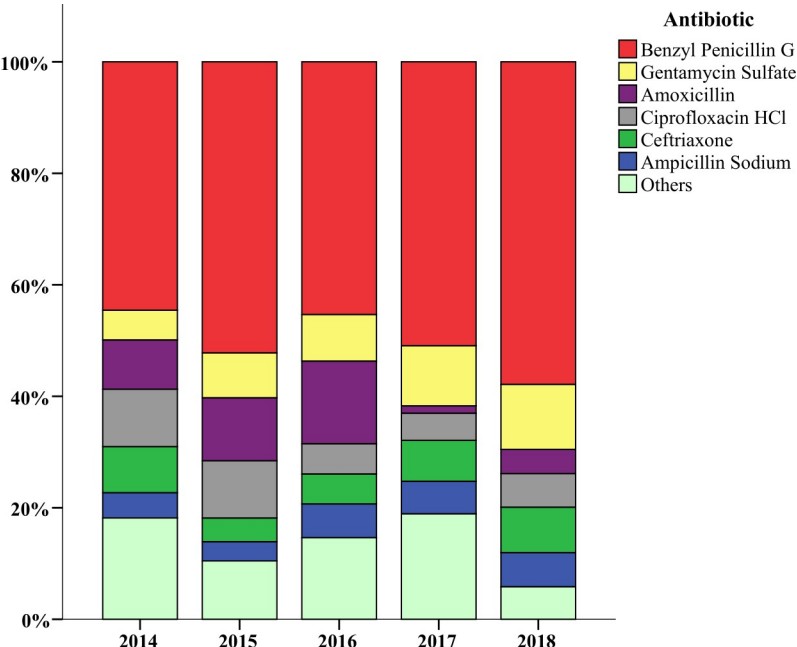

**Fig 3. Most commonly consumed antibiotics and their proportions Orotta and Hazhaz medical wards from 2014 to 2018 given in percentage.** Others: Doxcycline, co-trimoxazole, clarithromycin, flucloxacillin, metronidazole, chloramphenicol, cloxacillin and streptomycin.

HMW, however, increased from 139.54 to 164.48 DDD/100-BD. Our results suggest that the increased amount of antibiotic consumption in HMW is one form of irrational use of antibiotics seen in levels lower than tertiary care hospitals.

The three most commonly prescribed antibiotics in the world are penicillins, third-generation cephalosporins, and fluoroquinolones [28]. Our study shows that benzyl penicillin G was the most commonly consumed antibiotic in both OMW and HMW with a five-year mean average consumption of 35.4 DDD and 87.8 per DDD/100-BD respectively (Table 3). This is a similar finding in which penicillins were the most commonly consumed antibiotics in a secondary hospital from New Zealand [26]. The OMW consumption of benzyl penicillin G was higher than 17.88 DDD/100-BD in Nepal [29].We also found that oral antibiotics were less commonly consumed than parenteral antibiotics (Fig 2). Parenteral route of administration is more common and intravenous antibiotics make up 70% of total use in hospitals [30]. A 9 year study from inpatient Hospitals in Kazakhstan reports that oral antibiotics constituted from 35% to 58% even surpassing parenteral antibiotics in one year [31] while in our study we found that the use of oral antibiotic decreased across the study years (Fig 3).

Ciprofloxacin was the most commonly prescribed antibiotics in OMW with a five year mean 7.6 DDD/100-BD (Table 3), this is lower than the 3.42 DDD/100-BD reported from Nepal [29]. Ciprofloxacin is under the Watch Group of the AWaRe classification of the WHO. Watch group antibiotics could still be recommended as first line or second line treatment choices despite having a higher resistance potential, however, WHO recommends that these medicines be prioritized in local and national stewardship programs [32].

After benzyl penicillin G, gentamycin and amoxicillin with 17.7 and 16.9 DDD/100-BD respectively were the most commonly prescribed antibiotics. In HMW gentamycin is often used as an empirical treatment owing to its broad spectrum activity, alongside benzyl penicillin G or ampicillin; especially for the treatment of lower respiratory tract infections and infective

**Table 2. The number of beds, bed-days and occupancy rates of OMW and HMW from 2014–2018.**

| | 2014 | | 2015 | | 2016 | | 2017 | | 2018 | |
|---|---|---|---|---|---|---|---|---|---|---|
| | OMW | HMW | OMW | HMW | OMW | HMW | OMW | HMW | OMW | HMW |
| **No of beds** | 35 | 29 | 35 | 29 | 35 | 30 | 60 | 30 | 86 | 30 |
| **Occupancy rate** | 0.65 | 0.545 | 0.92 | 0.612 | 0.906 | 0.6 | 0.797 | 0.545 | 0.972 | 0.655 |
| **Bed-days** | 8,304 | 5,769 | 11,753 | 6,478 | 11,574 | 6,570 | 17,454 | 5,968 | 30,511 | 7,172.25 |

[1] OMW = Orotta Medical Ward, HMW = Hazhaz Medical Ward.

endocarditis. It is also used as add-on antibiotic in case of pneumonia management, when patients don't respond to benzyl penicillin G treatment alone. The consumption of amoxicillin in OMW is 4.3 DDD/100-BD, much lower than the 16.9 DDD/100-BD in HMW. This variation in consumption of oral amoxicillin may be due to the difference in the setting, one being tertiary hospital which have complicated and seriously ill patients' resulting in its low utilization, physicians opting for its parenteral forms for inpatient use. Oral amoxicillin is a preferred discharge medicine, i.e. given to patients at discharge time to continue to take it at home.

Third generation cephalosporins are thought to be medicines that are frequently consumed in hospitals [33, 34] and they are under the Watch group of the AWaRe classification [20]. In this study ceftriaxone was the third most commonly prescribed antibiotic in OMW (6 DDD/100-BD), higher than in Nepal 4.56 DDD/100-BD [29] but much lower than the 30 DDD/100-BD observed in an Ethiopian hospital [27], where cephalosporins were the most commonly used antibiotics. Ceftriaxone is a last resort antibiotic in Eritrean health care system, since it is the only cephalosporin available. Thus ceftriaxone requires extra prudence in its use; inappropriate and continuous use of ceftriaxone as empirical treatment would lead to inevitable serious antibiotic resistance emergence in these hospitals.

**Table 3. Antibiotic consumption in DDD per 100 bed-days in Orotta and Hazhaz medical wards.**

| | 2014 | | 2015 | | 2016 | | 2017 | | 2018 | | Mean (SD) | |
|---|---|---|---|---|---|---|---|---|---|---|---|---|
| | OMW | HMW | OMW | HMW | OMW | HMW | OMW | HMW | OMW | HMW | OMW | HMW |
| Benzyl Penicillin G | 37.34 | 75.12 | 46.44 | 108.06 | 48.24 | 69.87 | 22.44 | 84.48 | 22.40 | 101.48 | 35.4 (12.5) | 87.8 (14.7) |
| Amoxicillin | 7.83 | 14.45 | 7.66 | 25.73 | 6.19 | 51.55 | 0 | 2.79 | 0 | 9.3 | 4.3 (4.0) | 16.9 (10.8) |
| Gentamycin Sulfate | 3.89 | 9.59 | 3.15 | 20.69 | 4.98 | 16.77 | 4.20 | 18.43 | 2.08 | 22.87 | 3.7 (1.1) | 17.7 (4.5) |
| Ciprofloxacin HCl | 8.67 | 17.3 | 7.23 | 23.16 | 10.39 | 3.74 | 6.01 | 4.19 | 5.94 | 6.97 | 7.6 (1.9) | 11.1 (7.8) |
| Ceftriaxone | 7.78 | 13.18 | 2.92 | 9.66 | 8.06 | 5.91 | 4.96 | 10.47 | 6.30 | 11.15 | 6.0 (2.1) | 10.1 (2.4) |
| Ampicillin Sodium | 5.07 | 6.36 | 2.34 | 7.79 | 7.52 | 8.23 | 4.92 | 7.33 | 4.50 | 8.6 | 4.9 (1.8) | 7.7 (0.8) |
| Doxcycline | 5.28 | 0† | 3.83 | 0 | 6.05 | 0 | 6.87 | 16.76 | 0 | 0 | 4.4 (2.7) | 3.4 (6.7) |
| Cloxacillin Sodium | 2.49 | 1.3 | 1.77 | 0.39 | 3.05 | 3.93 | 3.42 | 1.26 | 0.62 | 2.27 | 2.3 (1.1) | 1.8 (1.2) |
| Metronidazole | 2.40 | 1.64 | 1.45 | 1.03 | 2.72 | 1.5 | 0.40 | 0.65 | 2.38 | 1.84 | 1.9 (0.9) | 1.3 (0.4) |
| Chloramphenicol | 0.65 | 0.6 | 0 | 0.51 | 0.71 | 2.94 | 0.43 | 0.14 | 0.04 | 0 | 0.4 (0.3) | 0.8 (1.1) |
| Flucloxacillin | 0.54 | 0 | 1.06 | 0 | 0 | 1.12 | 0.00 | 0 | 0 | 0 | 0.3 (0.5) | 0.2 (0.4) |
| Clarithromycin* | 2.30 | 0 | 4.25 | 0 | 0 | 0 | 0.00 | 0 | 0 | 0 | 1.3 (1.9) | 0 |
| Co-trimoxazole* | 12.10 | 0 | 4.36 | 0 | 6.05 | 0 | 3.87 | 0 | 2.13 | 0 | 5.7 (3.8) | 0 |
| Streptomycin* | 1.08 | 0 | 1.86 | 0 | 1.99 | 0 | 1.00 | 0 | 0.52 | 0 | 1.3 (0.6) | 0 |

[1] SD = Standard deviation; OMW = Orotta Medical Ward; HMW = Hazhaz Medical ward

* these antibiotics were not used in HMW

† no drug was dispensed.

A number of fluctuations in antibiotic consumption were seen over the five year period and this could be due to the availability of the medicines in the country, and might not reflect a true picture of how they would have been consumed had their availability been consistent. Yet, one objective of medicine consumption studies is to identify inconsistencies in medicine supply, and demonstrate how consumption of medicines are drastically affected by the availability of medicines on the market, which in turn leads to overuse of readily available broad spectrum antibiotics [35–37].

The findings of this study can be used as a starting point in the implementation and strengthening of Antimicrobial Stewardship Programs (ASPs) and encourage other hospitals throughout the country to measure their antibiotic consumption rates. Antibiotic consumption measurement and reporting are an important part of ASP. Many studies [34, 35, 38, 39] recommend implementation of ASP in hospitals to combat antimicrobial resistance due to over-consumption of antibiotics, since implementation of ASP is shown to reduce antibiotic consumption rates. The government should encourage the establishment of a sustainable antibiotic surveillance reporting framework, starting with the two national referral hospitals.

This study used the amount of antibiotics dispensed to medical wards to estimate the consumption rates of antibiotics and this might not always reflect the actual amount "consumed" by patients. However, even with this inherent limitation of the study, we believe the methodology adopted would be more than adequate to give a clue to the antibiotic consumption rate using a standardized metric. Future studies should focus on computing antibiotic consumption from all wards and use days-of-therapy to calculate antibiotic consumptions.

## Conclusions

This study measured antibiotic consumption trends in medical wards of two hospitals using DDD/100-BD methodology. The total antibiotic consumption in the HMW was almost double of that OMW. The trend analysis showed that antibiotic consumption in OMW was significantly decreased from 2014 to 2018 while there was no significant difference in consumption in HMW. Benzyl penicillin was the most consumed antibiotic in HMW and OMW throughout the study period at 87.8DDD/100-BD and 35.4 DDD/100-BD respectively. Ceftriaxone and ciprofloxacin were among the most commonly consumed antibiotics in both wards, given these two antibiotics are under the Watch Group they must closely be observed by establishing an ASP in their respective hospitals. Further studies which focus on all the wards and across different regions need to be conducted to generate a complete picture of antibiotic consumption patterns across the country, which would be instrumental in setting up a national antibiotic consumption baseline.

## Supporting information

**S1 File. Antibiotics consumption data of OMW and HMW 2014–2018.**
(XLSX)

## Acknowledgments

We would like to thank the director of medical store of ONRTH and HZRH medical store members for their assistance in helping us get the consumption data.

## Author Contributions

**Conceptualization:** Nebyu Daniel Amaha.

**Data curation:** Nebyu Daniel Amaha, Dawit G. Weldemariam.

**Formal analysis:** Nebyu Daniel Amaha.

**Investigation:** Dawit G. Weldemariam.

**Methodology:** Nebyu Daniel Amaha, Dawit G. Weldemariam.

**Visualization:** Nebyu Daniel Amaha.

**Writing – original draft:** Nebyu Daniel Amaha, Dawit G. Weldemariam, Yohana H. Berhe.

**Writing – review & editing:** Nebyu Daniel Amaha, Dawit G. Weldemariam, Yohana H. Berhe.

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
