## [Decision Letter · Decision Letter 0]

4 May 2020

Antibiotic consumption study in two hospitals in Asmara from 2014 to 2018 using WHO’s Defined Daily Dose (DDD) methodology

PONE-D-20-06944

Dear Dr. Amaha,

We are pleased to inform you that your manuscript has been judged scientifically suitable for publication and will be formally accepted for publication once it complies with all outstanding technical requirements.

With kind regards,

Iddya Karunasagar

Academic Editor

PLOS ONE

Journal Requirements:

1. Thank you for stating the following in the Financial Disclosure section: "The author(s) received no specific funding for this work."

We note that one or more of the authors are employed by a commercial company: 'Pharmacy, Addis Ababa, Ethiopia'.

Please respond by return email with an updated Funding Statement and Competing Interests Statement and we will change the online submission form on your behalf.

Additional Editor Comments (optional):

Though data is limited, this could form baseline information to develop antibiotic usage policy.

Reviewers' comments:

Reviewer's Responses to Questions

**Comments to the Author**

1. Is the manuscript technically sound, and do the data support the conclusions?

Reviewer #1: Yes

2. Has the statistical analysis been performed appropriately and rigorously? 

Reviewer #1: Yes

3. Have the authors made all data underlying the findings in their manuscript fully available?

Reviewer #1: Yes

4. Is the manuscript presented in an intelligible fashion and written in standard English?

Reviewer #1: Yes

5. Review Comments to the Author

Reviewer #1: While this is very relevant topic, drug utilization data presented in DDDs only give a rough estimate of consumption and not an exact picture of actual use. Although it enables the researchers to assess trends of antibiotic comsumption and to perform comparisons. Despite these limitations, this study can povide baseline information on the recent status of antibiotic overuse or misuse in two hospitals in Asmara. I hope the authors will succeed in developing monitoring system for antibiotic use.

6. PLOS authors have the option to publish the peer review history of their article (what does this mean?). If published, this will include your full peer review and any attached files.

Reviewer #1: No